# Feeding rates in sessile versus motile ciliates are hydrodynamically equivalent

Jingyi Liu[1], Yi Man[2], John H Costello[3,4], Eva Kanso[1,5]*

[1]Department of Aerospace and Mechanical Engineering, University of Southern California, Los Angeles, United States; [2]Mechanics and Engineering Science, Peking University, Beijing, China; [3]Department of Biology, Providence College, Providence, United States; [4]Whitman Center, Marine Biological Laboratories, Woods Hole, United States; [5]Department of Physics and Astronomy, University of Southern California, Los Angeles, United States

## eLife Assessment

This **important** paper addresses the role of fluid flows in nutrient uptake by microorganisms propelled by the action of cilia or flagella. Using a range of mathematical models for the flows created by such appendages, the authors provide **convincing** evidence that the two strategies of swimming and sessile motion can be competitive. These results will have significant implications for our understanding of the evolution of multicellularity in its various forms.

*For correspondence:
kanso@usc.edu

Competing interest: The authors declare that no competing interests exist.

**Abstract** Motility endows microorganisms with the ability to swim to nutrient-rich environments, but many species are sessile. Existing hydrodynamic arguments in support of either strategy, to swim or to attach and generate feeding currents, are often built on a limited set of experimental or modeling assumptions. Here, to assess the hydrodynamics of these 'swim' or 'stay' strategies, we propose a comprehensive methodology that combines mechanistic modeling with a survey of published shape and flow data in ciliates. Model predictions and empirical observations show small variations in feeding rates in favor of either motile or sessile cells. Case-specific variations notwithstanding, our overarching analysis shows that flow physics imposes no constraint on the feeding rates that are achievable by the swimming versus sessile strategies – they can both be equally competitive in transporting nutrients and wastes to and from the cell surface within flow regimes typically experienced by ciliates. Our findings help resolve a long-standing dilemma of which strategy is hydrodynamically optimal and explain patterns occurring in natural communities that alternate between free swimming and temporary attachments. Importantly, our findings indicate that the evolutionary pressures that shaped these strategies acted in concert with, not against, flow physics.

## Introduction

The dense and soluble nature of water allows nutrients necessary for survival to surround small organisms living in both fresh and marine ecosystems (*Barnes and Hughes, 1999*). However, the acquisition of these nutrients, either dissolved or particulate, is often challenging because they are frequently dilute or located within sparsely distributed patches (*Fenchel and Blackburn, 1999*; *Long and Azam, 2001*; *Durham et al., 2013*; *Keegstra et al., 2022*). Small, single-celled protists near the base of aquatic food chains have faced an evolutionary choice: either swim and use flows generated by swimming to encounter prey, or attach to a substrate and generate feeding currents from which to extract passing particles. Both 'swim' or 'stay' solutions occur among species in natural communities (*Zehr*

**eLife digest** Microorganisms living in aquatic ecosystems often face challenges in acquiring nutrients because resources are frequently diluted or unevenly distributed. To overcome these obstacles, organisms either swim toward nutrient-rich areas or attach to surfaces and generate feeding currents that draw in nutrients. However, research has long been inconclusive on which strategy is more efficient.

These 'swim' or 'stay' strategies shape material transport through aquatic trophic systems, affecting both individual fitness and broader processes such as global biogeochemical cycles and food web dynamics. Understanding microbial behavior is therefore essential for explaining patterns observed in natural communities and for clarifying mechanisms of material transport, nutrient acquisition, and the ecological roles of microorganisms.

Liu et al. examined how well these strategies work by analysing the shapes and flow patterns of ciliates. They found that both swimming and remaining attached can be equally effective for ciliates in typical aquatic environments. By combining mechanistic modelling with data from previous studies, they demonstrated that both strategies achieve comparable transport of nutrients and waste products. These findings suggest that evolutionary pressures have shaped these behaviours in alignment with the underlying physics of fluid flow.

They also showed that distributing ciliary activity across the cell surface improved the nutrient uptake by thinning the nutrient-depletion boundary layer – the region of fluid surrounding a cell where nutrient concentrations are reduced due to consumption. A thinner boundary layer allows nutrients to diffuse more rapidly toward the organism, revealing key design principles for maximising nutrient uptake.

Overall, Liu et al. resolve a longstanding debate by demonstrating that both feeding strategies of ciliates are equally effective. Their work provides a deeper insight into the behaviour of microorganisms and highlights the interplay between evolution and fluid mechanics. Future research should explore the evolutionary, ecological, and behavioural factors influencing feeding strategies, as well as the functional advantages of optimised cilia arrangements.

*et al., 2017*; *Barnes and Hughes, 1999*) and a number of species actively alternate between swimming and attachment (*Echigoya et al., 2022*). *Figure 1A* presents a focused survey of these strategies within a pivotal clade of microorganisms, the Ciliophora.

The 'swim' or 'stay' strategies shape material transport through this essential link in aquatic trophic systems, thus affecting not only the fitness of these microorganisms (*Stephens and Krebs, 1987*; *Rusconi and Stocker, 2015*; *Kanso et al., 2021*; *Krishnamurthy et al., 2023*) but also impacting global biogeochemical cycles and the food web chain (*Arrigo, 2005*; *Falkowski et al., 2008*; *Tréguer et al., 2018*; *Raina et al., 2022*). Therefore, understanding the flow physics underlying the exchange of nutrients and wastes at this scale is important across disparate fields of the life sciences, from evolutionary biology (*Solari et al., 2006*; *Shekhar et al., 2023*) to ecosystem ecology (*Guasto et al., 2012*; *Gasol and Kirchman, 2018*).

It has been generally appreciated that microorganisms, swimming or tethered, manipulate the fluid environment to maintain a sufficient turnover rate of nutrients and metabolites, unattainable by diffusive transport alone (*Karp-Boss et al., 1996*; *Solari et al., 2006*; *Pepper et al., 2010*; *Kanso et al., 2021*). However, to date, and with ample experimental (*Christensen-Dalsgaard and Fenchel, 2003*; *Jonsson et al., 2004*) and computational (*Michelin and Lauga, 2010*; *Andersen and Kiørboe, 2020*; *Klimenko et al., 2021*) studies, flow analysis has yielded contradictory results favoring either of the 'swim' (*Michelin and Lauga, 2010*; *Kirkegaard and Goldstein, 2016*; *Nguyen et al., 2019*; *Andersen and Kiørboe, 2020*) or 'stay' (*Christensen-Dalsgaard and Fenchel, 2003*; *Jonsson et al., 2004*) alternatives as optimal nutritional strategies.

If consideration of flow physics clearly favors one of the 'swim' (*Andersen and Kiørboe, 2020*) or 'stay' (*Christensen-Dalsgaard and Fenchel, 2003*) alternatives, then the existence of both indicates that the evolutionary pressures that led to the abundance of the other strategy had to act against flow physics and the propensity to optimize material transport to and from the cell surface. It would also imply that both solutions cannot occupy the same ecological niche without one of them being

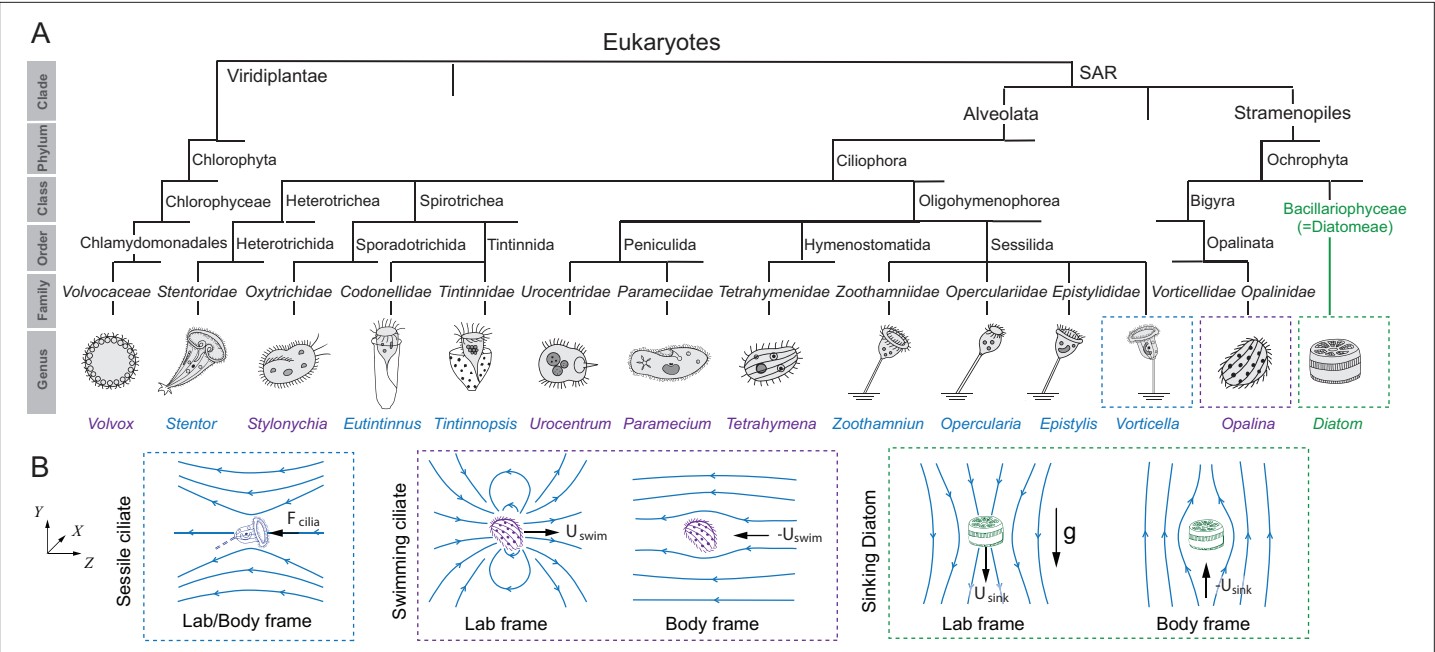

**Figure 1.** Phylogenetic tree. (**A**) Phylogenetic tree showing microorganisms known to feature cilia that generate feeding currents in either sessile (blue) or free swimming (purple) states. The class of diatoms – non-motile cells that sink when experiencing nutrient limitation – is shown for comparison. (**B**) Flow fields around a sessile ciliate, swimming ciliate, and sinking diatom, in lab and body frame of references. Streamlines are shown in blue in the lab frame ($X, Y, Z$).

seriously disadvantaged. An alternative possibility is that flow physics supports both solutions equally and that the choice of strategy does not compromise material transport to and from the cell surface. For example, in organisms that alternate between swimming and attachment, this transition is often influenced by external environmental conditions, such as pH balance (*Baufer et al., 1999*), nutrient concentration (*Langlois, 1975*) and prey availability (*Tartar, 2013*), and predator presence (*Dexter et al., 2019*).

But how can we distinguish between these two hypotheses? Establishing such a distinction is challenging because any attempt at quantifying flows around a specific microorganism (*Christensen-Dalsgaard and Fenchel, 2003*; *Catton et al., 2007*) inherently accounts for all evolutionary variables that shaped that microorganism and thus fails to provide a general and unbiased mechanistic understanding of the role of flow physics. Mathematical models allow objective comparison of the feeding rates achievable in the attached versus swimming states, while keeping all other variables the same. Surprisingly, besides (*Andersen and Kiørboe, 2020*), there is a paucity of mathematical studies that directly address this question. Importantly, results based on any single model naturally depend on the modeling assumptions; thus, any attempt at drawing general conclusions from considering a single organism or mathematical model should be carefully scrutinized.

In this study, we propose a systematic approach to address existing limitations in evaluating the hydrodynamics of the 'swim' or 'stay' alternatives. Our approach combines a survey of existing experimental observations within the entire Ciliophora clade (*Figure 1*) with mathematical models that span the morphology and flow conditions within which all surveyed ciliates fall (*Figure 2*). We additionally include a comparison with diatoms to distinguish the effects of relative body motion independent of cilia-driven feeding currents. We find, based on both empirical observations and mathematical models, that encounter rates of the swim and stay strategies converge under realistic conditions and are essentially equivalent within flow regimes typically experienced by ciliates.

## Results

Our results are organized around three main themes: (A) comparative analysis of morphologies, size, and fluid flows in sessile and swimming ciliates and sinking diatoms, (B) evaluation of nutrient

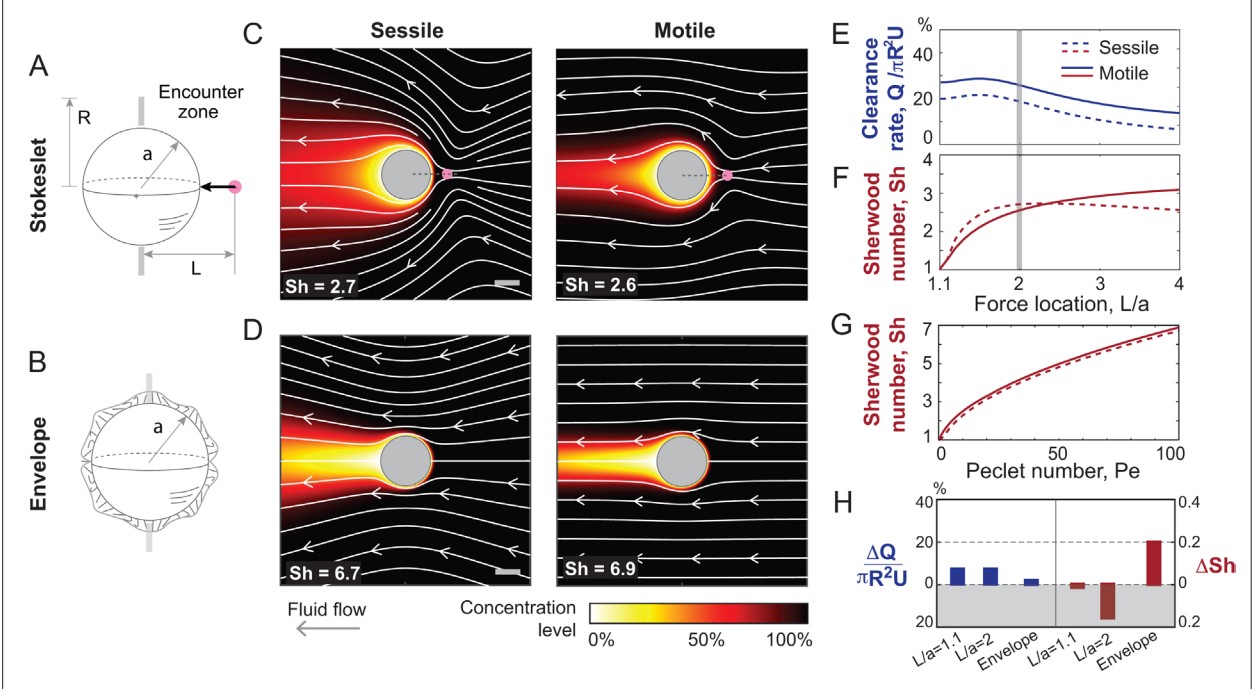

**Figure 2.** Stokeslet and envelope models of sessile and motile ciliates. (**A**) Stokeslet model where ciliary activity is represented by a Stokeslet force $F_{cilia}$ is located at a distance $(L - a)/a$ outside the spherical cell surface with no-slip surface velocity. (**B**) Envelope model where cilia activity is distributed over the entire cell surface with slip surface velocity. (**C, D**) Fluid streamlines (white) and nutrient concentration fields (colormap) in the sessile and swimming cases. Here, $L/a = 2$, $a = 1$ and $F_{cilia}$ is chosen to generate a swimming speed $U = 2/3$ in the motile case to ensure consistency with the envelope model. (**E, F**) Nutrient uptake in sessile and motile Stokeslet-sphere model based on calculation of clearance rate $Q$ of a fluid volume passing through an annular disk of radius $R/a = 1.1$ and Sherwood number Sh. In the latter, Pe is 100. (**G**) Nutrient uptake in sessile and motile envelope model based on calculation of Sherwood number Sh as a function of Pe. (**H**) Difference in clearance rate $\Delta Q = Q_{motile} - Q_{sessile}$ and Sherwood number $\Delta Sh = \Delta I/I_{diffusion} = Sh_{motile} - Sh_{sessile}$ in the Stokeslet-sphere model for $L/a = 1.1$ and $L/a = 2$ and in the envelope model. In both metrics, the difference is less than 20%: $\Delta Q$ is less than 20% the advective flux $\pi R^2 U$ and $\Delta I$ is less than 20% of the corresponding diffusive uptake $I_{diffusion} = 4\pi RDC_\infty$. The shaded gray area denotes when the sessile strategy is advantageous.

uptake in mathematical models of sessile and motile feeders spanning the Stokeslet (*Pepper et al., 2013*; *Kanso et al., 2021*; *Andersen and Kiørboe, 2020*) and envelope (*Blake, 1971b*; *Michelin and Lauga, 2011*) models and covering the entire range within which all surveyed ciliates fall (*Omori et al., 2020*), (C) analysis of biological data in light of model prediction and of asymptotic analysis in the two extremes of diffusion and advection dominant limits.

## Comparative morphometric, phylogenetic, and flow data in ciliates and diatoms

We conducted a survey on the morphology, flows, and phylogenetic lineage of ciliates and diatoms (*Keeling et al., 2005*; *Parfrey et al., 2008*; *Hinchliff et al., 2015*; *Grattepanche et al., 2018*; *Figure 1*).

Sessile ciliates, such as the *Stentor* (*Wan et al., 2020*), *Opercularia* (*Sládecek, 1981*; *Zima-Kulisiewicz and Delgado, 2009*), and *Vorticella* (*Sleigh and Barlow, 1976*; *Noland and Finley, 1931*; *Vopel et al., 2002*; *Nagai et al., 2009*; *Pepper et al., 2013*), are characterized by a ciliary crown, where the motion of beating cilia entrains fluid toward the cell. The cell body and ciliary crown are positioned away from the surface they live upon, usually with a stalk, to minimize the effect of that surface on slowing down the cilia-driven microcurrents (*Laybourn, 1976*; *Boenigk and Arndt, 2002*; *Pepper et al., 2010*; *Pepper et al., 2013*). Furthermore, to avoid generating recirculating microcurrents and reduce reprocessing of depleted water (*Sládecek, 1981*; *Vopel et al., 2002*; *Pettitt et al., 2002*), sessile ciliates actively regulate their orientation to feed at an angle relative to the substratum (*Pepper et al., 2013*). At optimal inclination, the effective cilia-generated force is nearly parallel to the bounding substrate and creates quasi-unidirectional flows that drive nutrients and particles past

**Table 1.** Survey of size $a$ and flow measurements $U$ in sessile and swimming ciliates and sinking diatoms.

Size $a$ is calculated using the volume-equivalent spherical radius. Corresponding ranges of Pe numbers are based on the diffusivity of oxygen, $D = 10^{-9} \text{m}^2 \cdot \text{s}^{-1}$, live bacteria, $D = 4 \times 10^{-10} \text{m}^2 \cdot \text{s}^{-1}$,, and dead bacteria $D = 2 \times 10^{-13} \text{m}^2 \cdot \text{s}^{-1}$.

| | Empirical measurements | | Péclet number, Pe | | |
| --- | --- | --- | --- | --- | --- |
| | microorganism size | characteristic speed | oxygen diffusivity | live bacteria diffusivity | dead bacteria diffusivity |
| Sessile ciliates | 15–60 | 50–2500 | 1–80 | 2–210 | $(5–400) \times 10^3$ |
| Swimming ciliates | 15–180 | 50–3200 | 1–160 | 8–390 | $(17–800) \times 10^3$ |
| Sinking diatoms | 10–120 | 40–210 | 0.4–23 | – | – |

the cell feeding apparatus (*Pepper et al., 2013*; *Wandel and Holzman, 2022*; *Kiørboe, 2024*). In motile ciliates, such as the *Paramecium* and *Volvox*, the surface of the organism is often entirely covered with cilia that beat in a coordinated manner and power the organism to swim through the surrounding fluid (*Bullington, 1930*; *Brennen and Winet, 1977*; *Lauga and Powers, 2009*; *Lisicki et al., 2019*). Diatoms lack motility apparatus and sink by regulating their buoyancy (*Karp-Boss et al., 1996*; *Miklasz and Denny, 2010*; *Gemmell et al., 2016*; *Figure 1B*).

Empirical flow measurements around sessile (*Pepper et al., 2010*; *Pepper et al., 2021*; *Zima-Kulisiewicz and Delgado, 2009*; *Hartmann et al., 2007*; *Nagai et al., 2009*; *Wandel and Holzman, 2022*) and motile (*Emlet, 1990*; *Drescher et al., 2010*) ciliates are sparse. Here, we collected morpho-metric and flow data from published work covering ten species of sessile ciliates (*Vopel et al., 2002*; *Sleigh and Barlow, 1976*; *Sládecek, 1981*; *Nagai et al., 2009*; *Pepper et al., 2013*; *Wan et al., 2020*), ten species of swimming ciliates (*Bullington, 1930*; *Brennen and Winet, 1977*; *Lisicki et al., 2019*), and seven species of diatoms (*Miklasz and Denny, 2010*; *Gemmell et al., 2016*). A summary of the ranges of sizes and characteristic speeds are reported in *Table 1* and *Appendix 1—figure 1*; detailed measurements are listed in a supplemental data file. Size is represented by the volume-equivalent spherical radius $a$ (*Appendix 1—figure 2*). The characteristic speeds $U$ for sessile ciliates are based on the maximal flow speeds measured near the ciliary crown. For swimming ciliates and sinking diatoms, we collected and measured swimming and sinking speeds, which, given the no-slip boundary condition in this viscous regime (*Purcell, 1977*; *Lauga and Powers, 2009*), also represent flow speeds near the surface of these microorganisms.

Phylogenetically, all surveyed microorganisms, except the *Volvox*, belong to the SAR supergroup, encompassing the Stramenopiles, Alveolates, and Rhizaria clades (*Figure 1*). The Rhizaria clade is not represented in our survey because it mostly consists of ameboids, while its flagellates have complex and functionally ambiguous morphologies that do not fit in the present analysis (*Keeling et al., 2005*; *Grattepanche et al., 2018*). *Volvox*, the only multicellular microorganism listed in *Figure 1*, is an algae that belongs to the Viridiplantae clade. Diatoms evolved from the same SAR supergroup as the majority of unicellular ciliates, but without the ciliary motility apparatus, and while early ciliates date back to about 700 million years (*Bosak et al., 2011*), diatoms appeared later, about 200 million years ago (*Sorhannus, 2007*; *Medlin, 2011*; *Nakov et al., 2018*). Diatoms generally exist in a suspended state and sink under low nutrient conditions (*Bienfang et al., 1982*; *Karp-Boss et al., 1996*; *Gemmell et al., 2016*). Of the twelve ciliates listed in *Figure 1*, many transition during their lifecycle between sessile and free-swimming states (*Jonsson et al., 2004*; *Bickel et al., 2012*). *Stentors* become rounder when swimming (*Slabodnick et al., 2017*).

The microcurrents generated by these ciliates improve solute transport to and from the surface of the microorganism. For a characteristic microcurrent of speed $U = 100\ \mu\text{m} \cdot \text{s}^{-1}$, small molecules and particles would be transported over a characteristic distance $a = 100$ µm in approximately $a/U = 1$ s. In contrast, the same substance transported by diffusion alone takes a considerably longer time to traverse the same distance. For example, diffusive transport of oxygen and small molecules, with diffusivities that are in the order of $D = 10^{-9} \text{m}^2 \cdot \text{s}^{-1}$, takes about $a^2/D = 10$ s, while live and dead bacterial particles with respective diffusivity $D = 4 \times 10^{-10} \text{m}^2 \cdot \text{s}^{-1}$ and $D = 2 \times 10^{-13} \text{m}^2 \cdot \text{s}^{-1}$ (*Berg, 2018*) take about $a^2/D = 25$ s and 10,000 s, respectively. The ratio of diffusive $a^2/D$ to advective $a/U$ timescales defines the Péclet number, $\text{Pe} = aU/D$. For $\text{Pe} \ll 1$, mass transport is controlled by

molecular diffusion. For the microorganisms that we surveyed, we obtained Pe ranging from nearly 0 to as large as $10^3$–$10^5$ depending on the nutrient diffusivity (*Table 1*). This dimensional analysis suggests that the flows generated by the microorganisms substantially enhance the transport of solutes to and from their surface, and while it clearly shows that diatoms typically occupy a smaller range of Pe numbers, this analysis does not reveal a clear distinction between sessile and swimming ciliates. To further explore such distinction, if present, and to assess whether ciliates are disadvantaged by flow physics in their attached state compared to their swimming state, as suggested in *Andersen and Kiørboe, 2020*, we developed mathematical models that allow for an unbiased comparison between these two states.

## Mathematical modeling of fluid flows and nutrient uptake

To quantify and compare nutrient uptake across microorganisms, we approximated the cell body by a sphere of radius $a$, as typically done in modeling sessile and swimming ciliates (*Blake, 1971b*; *Magar et al., 2003*; *Michelin and Lauga, 2011*; *Andersen and Kiørboe, 2020*) and sinking diatoms (*Riley, 1952*; *Karp-Boss et al., 1996*; *Kanso et al., 2021*; *Figure 2*).

The fluid velocity $\mathbf{u}$ around the sphere is governed by the incompressible Stokes equations, $-\nabla p + \eta\nabla^2\mathbf{u} = 0$ and $\nabla \cdot \mathbf{u} = 0$, where $p$ is the pressure field and $\eta$ is viscosity. We solved these equations in spherical coordinates $(r, \theta, \phi)$, considering axisymmetry in $\phi$ and proper boundary conditions. In the motile case, we solved for the fluid velocity field $\mathbf{u}$ in body frame by superimposing a uniform flow of speed $U$ equal to the swimming speed past the sphere; we calculated the value of $U$ from force balance considerations (*Dölger et al., 2017*; *Andersen and Kiørboe, 2020*) (see SI for details).

We solved the Stokes equations for two models of cilia activity: cilia represented as a Stokeslet force $F_{\text{cilia}}$ placed at a distance $L$ and pointing towards the center of the sphere and no-slip velocity at the spherical surface (*Blake, 1971a*; *Wróbel et al., 2016*; *Andersen and Kiørboe, 2020*; *Kim and Karrila, 1991*; *Figure 2A*), and densely packed cilia defining an envelope model with a slip velocity $\mathbf{u}|_{r=a} = \mathcal{U}\sin\theta$ at the spherical surface where all Cilia exert tangential forces pointing from one end of the sphere to the opposite end (*Blake, 1971b*; *Michelin and Lauga, 2010*; *Michelin and Lauga, 2011*; *Figure 2B*). Detailed expressions of the flow fields and governing equations in both models are included in the SI (*Appendix 1—tables 1 and 2*). In dimensionless form, we set the cell's length scale $a = 1$ and tangential velocity scale $\mathcal{U} = 1$ in the envelope model, and we set the ciliary force $F_{\text{cilia}}$ in the Stokeslet model to produce the same swimming speed ($U = 2/3$) as in the envelope model when the sphere is motile.

To evaluate the steady-state concentration of dissolved nutrients around the cell surface, we numerically solved the dimensionless advection-diffusion equation $\text{Pe}\,\mathbf{u} \cdot \nabla C = \Delta C$ in the context of the Stokeslet and envelope models. Here, the advective and diffusive rates of change of the nutrient concentration field $C$, normalized by its far-field value $C_\infty$, are given by $\text{Pe}\,\mathbf{u} \cdot \nabla C$ and $\Delta C$, respectively, with $\nabla C$ the concentration gradient. At the surface of the sphere, the concentration is set to zero to reflect that nutrient absorption at the surface of the microorganism greatly exceeds transport rates of molecular diffusion (*Berg and Purcell, 1977*; *Bialek, 2012*; *Short et al., 2006*).

In *Figure 2C and D*, flow streamlines (white) and concentration fields (colormap at Pe = 100) are shown in the Stokeslet and envelope models. In the sessile sphere, ciliary flows drive fresh nutrient concentration from the far field towards the ciliated surface. These fresh nutrients thin the concentration boundary layer at the leading surface of the sphere, where typically the cytostome or feeding apparatus is found in sessile ciliates, with a trailing plume or 'tail' of nutrient depletion. Similar concentration fields are obtained in the swimming case, albeit with a narrower trailing plume.

To assess the effects of these cilia-generated flows on the transport of nutrients to the cell surface, we used two common metrics of feeding. First, we quantified fluid flux or clearance rate $Q$ through an encounter zone near the organism's oral surface (*Christensen-Dalsgaard and Fenchel, 2003*; *Pepper et al., 2013*; *Shekhar et al., 2023*). Namely, following *Andersen and Kiørboe, 2020*, we defined the clearance rate $Q = -2\pi \int_a^R \mathbf{u} \cdot \mathbf{e}_z\big|_{z=0}\, r dr$, normalized by the advective flux $\pi R^2 U$, over an annular encounter zone of radius $R$ extending radially away from the cell surface (*Figure 2A*). Second, we quantified the concentration flux of dissolved nutrients at the cell surface (*Kanso et al., 2021*; *Michelin and Lauga, 2010*; *Michelin and Lauga, 2011*). To this end, we integrated the inward concentration flux $I = \int_S D\nabla C \cdot \hat{\mathbf{n}} dS$, normalized by the diffusive nutrient uptake $I_{\text{diffusion}} = 4\pi a D C_\infty$ to get the Sherwood number $\text{Sh} = I/I_{\text{diffusion}}$. We applied both metrics to each of the Stokeslet and envelope models.

In *Figure 2E*, we report the clearance rate $Q$ in the context of the Stokeslet model as a function of the ciliary force location $L/a$ for a small annular encounter zone of radius $R = 1.1a$ extending away from the cell surface. Swimming is always more beneficial. However, the increase in clearance rate due to swimming is less than 10%. This is in contrast to the several-fold advantage obtained in *Andersen and Kiørboe, 2020* for $L = 4a$ and $R = 10a$. (results of *Andersen and Kiørboe, 2020* are reproduced in Fig. S3). We employed the same metric $Q$ in the envelope model and found that motility is also more advantageous, albeit at less than 5% benefit (*Figure 2H*).

A few comments on the choice of the size of the encounter zone are in order. Nutrient encounter and feeding in ciliates occur near the leading edge of the ciliary band (*Gilmour, 1978*; *Thomazo et al., 2021*; *Jiang and Buskey, 2025a*; *Jiang and Buskey, 2025b*). Cilia are typically of the order of 10 microns in length, and the cell body of a ciliate is typically in the range of 10–1000 microns. We chose $R = 1.1a$ indicating encounter within an annular protrusion extending 10% beyond the body radius because it falls within the biological range and because a larger encounter zone would induce additional drag on the body that needs to be accounted for in the model. In contrast, *Andersen and Kiørboe, 2020* chose an encounter zone extending up to 900% the body radius, without accounting for the drag that such a large collection area would add to a swimming body. This also exceeds biological considerations in most ciliates and flagellates, even in *Choanoflagellates* (*Nielsen et al., 2017*) and *Chlamydomonas* (*Nielsen et al., 2017*), where the flagellum length could be up to six times the cell radius.

In *Figure 2F and G*, we report the Sh number based on the Stokelet and envelope models, respectively. In the Stokeslet model (*Figure 2F*), sessile spheres do better when the cilia force is close to the cell surface $(L − a)/a \lesssim 1.25$. In the envelope model (*Figure 2G*), motile spheres do slightly better for all Pe $\lesssim 100$. The difference $\Delta$Sh between the sessile and motile spheres favors, by less than 20%, the sessile strategy in the Stokeslet model and the swimming strategy in the envelope model (*Figure 2H*).

Comparing Sh between the Stokeslet and envelope models (*Figure 2C and D*), we found that, at Pe = 100, Sh = 2.7 (sessile) and 2.6 (motile) in the Stokeslet model compared to Sh = 6.7 (sessile) and 6.9 (motile) in the envelope model. This is over a twofold enhancement in nutrient uptake at the same swimming speed $U = 2/3$ simply by distributing the ciliary force over the entire surface of the cell! Indeed, this improvement occurs because the ciliary motion in the envelope model significantly thins

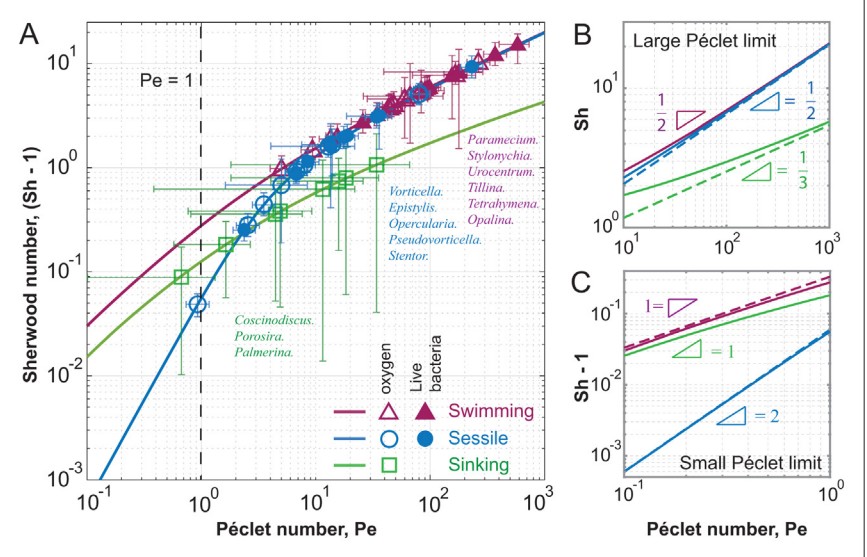

**Figure 3.** Sherwood number versus Péclet number for the sinking (green) diatom and the swimming (purple) and sessile (blue) ciliates based on the envelope model. (**A**) Shifted Sherwood number (Sh - 1) versus Péclet number in the logarithmic scale for a range of Pe from 0 to 1000. Pe numbers associated with experimental observations of diatoms (square), swimming ciliates (triangle), and sessile ciliates (circle) are superimposed. Corresponding Sh numbers are calculated based on the mathematical model. Empty symbols are for oxygen diffusivity $D = 1 \times 10^{-9} \text{m}^2 \cdot \text{s}^{-1}$ and the solid symbols correspond to the diffusivity $D = 4 \times 10^{-10} \text{m}^2 \cdot \text{s}^{-1}$ of live bacteria (*Berg, 2018*). (**B–C**) Asymptotic analysis (dashed lines) of Sherwood number in the large Péclet limit (**B**) and small Péclet limit (**C**).

the concentration boundary layer along the entire cell surface as opposed to only near where the cilia force is concentrated in the Stokeslet model.

In our survey of sessile and motile ciliates (*Figure 1*), cilia are clearly distributed over the cell surface. Thus, we next explored in the context of the envelope model the behavior of the Sh number across a range of Pe values that reflect empirical values experienced by the surveyed ciliates (*Table 1*).

## Linking model prediction to biological data

We numerically computed the Sherwood number for a range of $Pe \in [0, 1000]$ for the sessile and motile spheres, and, to complete this analysis, we calculated the Sh number around a sinking sphere. Numerical predictions (*Figure 3A*, solid lines, log-log scale) show that at small Pe, swimming is more advantageous than attachment; in fact, any motion, even sinking, is better than no motion at all (*Solari et al., 2006*). However, at larger Pe, there is no distinction in Sh number between the sessile and motile sphere.

We next used as input to the sessile, swimming, and sinking sphere models the Pe numbers obtained from experimental measurements of sessile (blue ◯) and swimming (purple △) ciliates and sinking diatoms (green ▢), respectively, and we computed the corresponding values of Sh number (*Figure 3A*). Sinking diatoms are characterized by smaller values of Sh number, whereas with increasing Pe, the Sh values of sessile ciliates span similar ranges as those of swimming ciliates.

To complete this analysis, we probed the feeding rates under extreme Péclet limits. We extended the asymptotic scaling analysis developed in *Acrivos and Taylor, 1962*; *Acrivos and Goddard, 1965* and translated to nutrient uptake in sinking diatoms (*Karp-Boss et al., 1996*) and swimming ciliates (*Magar et al., 2003*; *Michelin and Lauga, 2011*), to arrive at asymptotic expressions for sessile ciliates in the two limits of small and large Pe (*Appendix 1—table 3*),

$$Pe \ll 1 : Sh = 1 + \frac{43}{720}Pe^2, \quad Pe \gg 1 : Sh = \frac{2}{\sqrt{3\pi}}Pe^{\frac{1}{2}}.$$

In *Figure 3B and C*, we superimposed our asymptotic results, together with the asymptotic results of *Karp-Boss et al., 1996*; *Acrivos and Taylor, 1962*; *Acrivos and Goddard, 1965*; *Magar et al., 2003*; *Michelin and Lauga, 2011*, onto our numerical findings. At small $Pe \ll 1$, the Sh numbers for swimming and sinking spheres scale similarly with Pe ($Sh \sim Pe$), whereas Sh scales worse ($Sh \sim Pe^2$) for the sessile sphere. Our thorough literature survey indicates, save one, no data points for sessile microorganisms in this limit. At large $Pe \gg 1$, the Sh numbers of the sessile and swimming spheres scale similarly with Pe ($Sh \sim Pe^{\frac{1}{2}}$), whereas the sinking sphere scales worse ($Sh \sim Pe^{\frac{1}{3}}$). Similar scaling is found in swimming ciliate models (*Magar et al., 2003*; *Short et al., 2006*; *Michelin and Lauga, 2011*). These results confirm that, hydrodynamically, sessile and swimming ciliates are equivalent in the limit of large Pe. When cilia generate strong feeding currents that drive nutrients and particulates toward the cell body, attached microorganisms can be equally competitive with motile microorganisms that swim to feed.

## Discussion

We contributed a comprehensive methodology for evaluating the role of flow physics and comparing feeding rates in motile and sessile ciliates. Our approach combined a survey of previously published empirical measurements of ciliates' shape and velocity with two mechanistic models of cilia-driven flows (concentrated point force and distributed force density) and two metrics of nutrient uptakes (clearance rate and Sherwood number) in attached and swimming ciliates. The concentrated versus distributed ciliary force models form two extreme limits within which all surveyed ciliates fall. Clearance rate measures advective material transport through an encounter zone, which is independent of Pe; Sh number accounts for both diffusive and advective transport and varies with Pe.

The difference in feeding rates between the sessile and motile strategies depended on the choice of model, model parameters, and feeding metric (*Figure 2*).

In the context of the concentrated force model and considering clearance rate as a metric for feeding, we found that it is better to swim than to attach, but these advantages are modest (less than 20%) under justifiable conditions of ciliary force placement and encounter zone close to the cell

surface. In *Andersen and Kiørboe, 2020*, several-fold improvements were reported for swimming using the same model and feeding metric but questionable parameter values - clearance rates were computed through an encounter zone that extended up to ten body lengths away from the cell surface without accounting for the effect that such an extensive collection surface would have on drag generation during swimming (*Andersen and Kiørboe, 2020*). We showed that for a small encounter zone that justifies omission of these drag forces, the improvement in clearance rate during swimming is much smaller than predicted in *Andersen and Kiørboe, 2020*.

Surprisingly, using the same concentrated force model, we found that attachment improves nutrient uptake when considering concentration of dissolved nutrients at Pe = 100 and measuring the Sherwood number associated with nutrient uptake over the entire cell surface. Again, the improvement is modest (*Figure 2H*). Taken together, these results show that in the same model, two different feeding metrics favor different strategies, albeit at a slim advantage of less than 20% in favor of either swimming or attachment.

When distributing the ciliary force over the entire cell surface, we found, using either metric, that swimming is more beneficial by a very small margin for Pe ≤ 100 (*Figure 2G*). Interestingly, the difference in Sh number between swimming and attached cells decreases at larger Pe values (*Figure 3A*), and in the asymptotic limit of $Pe \gg 1$, Sh scales similarly with Pe ($Sh \propto Pe^{1/2}$) for both swimming and sessile cells (*Figure 3A and B*). That is, at large Pe, material transport to and from the cell surface is not compromised by the choice of strategy.

From our survey of previously published empirical measurements of ciliates' shape and velocity (*Figure 1*), we extracted biologically relevant ranges of Pe values (*Table 1*) and combined these empirical observations with model predictions (*Figure 3A*). We found significant overlap in Sh number between sessile and motile organisms at a wide range of representative Pe values. These findings clearly show that both attachment and free swimming can lead to similar nutrient acquisition within a wide range of flows and Péclet values typically experienced by ciliates.

This study provides a fresh perspective on evaluating the role of flow physics in the feeding strategies of microorganisms. Prior methods in support of either the motile or sessile strategies as optimal drew general conclusions from focused analyses. Support for swimming came principally from flow-based models of idealized organisms propelling themselves through water (*Michelin and Lauga, 2010*;

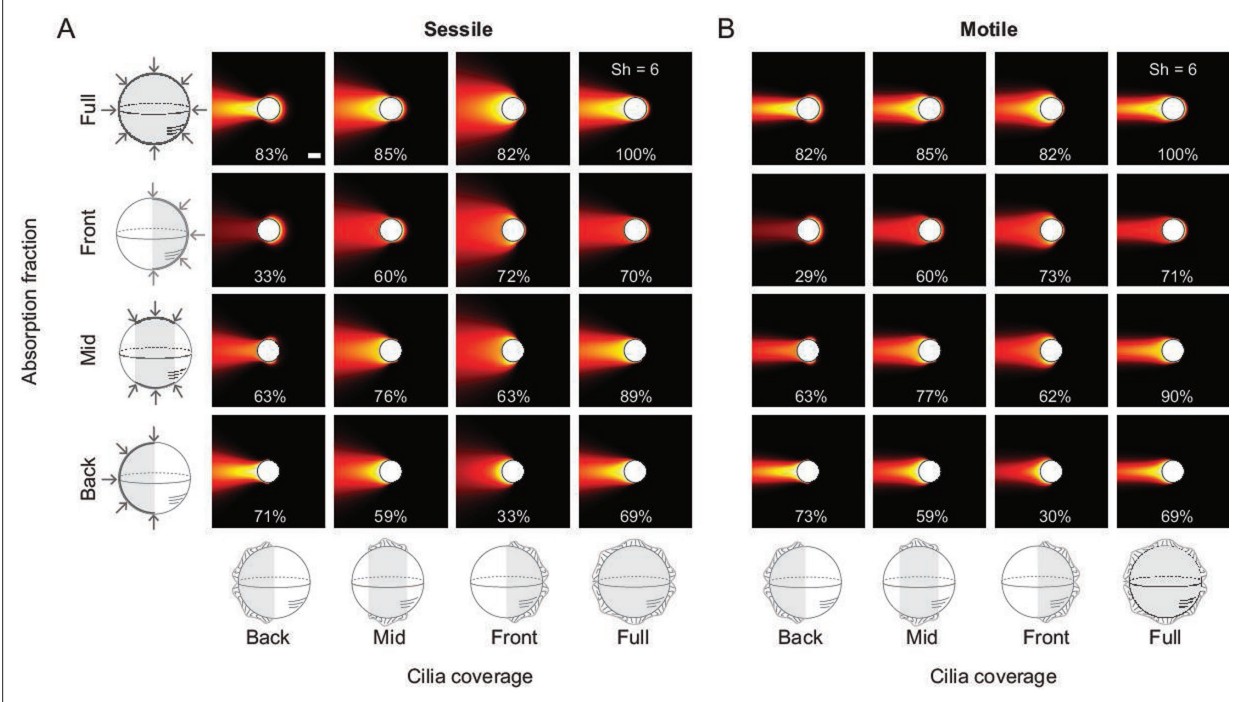

**Figure 4.** Robustness to variations in cilia coverage and absorption fraction. We considered a 50% cilia coverage and 50% absorption fraction located at back, middle, and front of the (**A**) sessile and (**B**) motile sphere. Concentration fields and Sherwood numbers with 100% cilia coverage and absorption area are shown in the top right corner. In all other cases, the Sh number is reported as a percentage of the full coverage/absorption case.

*Kirkegaard and Goldstein, 2016*; *Nguyen et al., 2019*; *Andersen and Kiørboe, 2020*). Support for maximum feeding by attached protists came from empirical measurements of prey removal by swimming versus attached individuals (*Jonsson et al., 2004*; *Christensen-Dalsgaard and Fenchel, 2003*). Our approach shows that, while feeding rates may vary between organisms and mathematical models, given a cellular (ciliary) machinery that allows microorganisms to manipulate the surrounding fluid and generate flows, flow physics itself imposes no constraint on what is achievable by the swimming versus sessile strategies – they can both be equally competitive in transporting nutrients and wastes to and from the cell surface in the large Pe limit where nutrient advection is dominant. Our findings suggest that the choice of feeding strategy was likely influenced by evolutionary, ecological, or behavioral variables other than flow physics, such as metabolic or sensory requirements (*Bezares-Calderón et al., 2020*; *Mitchell, 2007*; *Bloodgood, 2010*), predator avoidance (*Pierce and Turner, 1992*), symbiotic relations (*Kanso et al., 2021*), and nutrient availability or environmental turbulence (*Pierce and Turner, 1992*; *Lauro et al., 2009*; *Klimenko et al., 2021*).

Along with assessing feeding rates in motile versus sessile strategies, our analysis revealed interesting 'design' principles for maximizing nutrient uptake by distributing ciliary activity over the entire cell surface (*Figure 2*). This design thins the nutrient-depletion boundary layer at the surface of the cell where absorption occurs: for the same overall swimming speed, distributing ciliary activity over the cell surface improves nutrient uptake by over twofold compared to when the ciliary force is concentrated at one location (*Figure 2*). Indeed, cilia are often distributed over a portion or entire cell surface in sessile and motile ciliates, with some variability in cilia distribution and cell surface fraction where prey is intercepted (*Figure 1*). To account for such variability, we computed the flow and concentration fields under various perturbations to cilia coverage and surface fraction where absorption takes place (*Figure 4*). For each perturbation, we calculated the Sh number in the form of a percentage of that corresponding to full cilia coverage and absorption over the entire surface. We found small differences in Sh numbers between the sessile and motile spheres. Our findings – that the motile and sessile strategies are equivalent in terms of material transport to the cell surface – are thus robust to cilia perturbations. Additionally, we found that, for a given cilia coverage, nutrient uptake is maximized when the absorption surface coincides with the cilia coverage area. This design – cilia collocated with the cell feeding apparatus – is abundant in sessile protists (*Figure 1*). Our findings open new venues for investigating the functional advantages of optimal cilia designs (cilia number and distribution) that maximize not only locomotion performance (*Omori et al., 2020*) but also feeding rates and for evaluating the interplay between cell design and feeding strategies (sessile versus motile) both at the unicellular (*Liu et al., 2025*) and multicellular levels (*Day et al., 2022*). These future directions will enrich our understanding of the complexity of feeding strategies in ciliates and how strategy and design have evolved to provide behavioral advantages to these microbes.

## Acknowledgements

EK acknowledges support from the University of Southern California (for PhD student JL) and support from the Office of Naval Research (ONR) Grants N00014-22-1-2655, N00014-19-1-2035, N00014-17-1-2062, and N00014-14-1-0421; the National Science Foundation (NSF) Grants RAISE IOS-2034043, CBET-2100209, and INSPIRE MCB-1608744; the National Institutes of Health (NIH) Grant R01 HL 153622–01 A1; and the Army Research Office (ARO) Grant W911NF-16-1-0074.

## Additional information

### Funding

| Funder | Grant reference number | Author |
| --- | --- | --- |
| Office of Naval Research | N00014-22-1-2655 | Eva Kanso |
| U.S. National Science Foundation | IOS-2034043 | Eva Kanso |
| National Institutes of Health | R01 HL 153622-01A1 | Eva Kanso |

| Funder | Grant reference number | Author |
|---|---|---|
| United States Army Research Office | W911NF-16-1-0074 | Eva Kanso |
| Office of Naval Research | N00014-19-1-2035 | Eva Kanso |
| Office of Naval Research | N00014-14-1-0421 | Eva Kanso |
| Office of Naval Research | N00014-17-1-2062 | Eva Kanso |
| U.S. National Science Foundation | CBET-2100209 | Eva Kanso |
| U.S. National Science Foundation | INSPIRE MCB-1608744 | Eva Kanso |

The funders had no role in study design, data collection and interpretation, or the decision to submit the work for publication.

### Author contributions

Jingyi Liu, Data curation, Formal analysis, Validation, Investigation, Visualization, Methodology, Writing – original draft, Writing – review and editing; Yi Man, Methodology; John H Costello, Data curation, Validation, Methodology, Writing – review and editing; Eva Kanso, Conceptualization, Supervision, Funding acquisition, Investigation, Methodology, Writing – original draft, Writing – review and editing, Formal analysis

### Author ORCIDs

Jingyi Liu ⓘ https://orcid.org/0009-0007-7897-7742
Yi Man ⓘ https://orcid.org/0000-0002-6267-0216
John H Costello ⓘ https://orcid.org/0000-0002-6967-3145
Eva Kanso ⓘ https://orcid.org/0000-0003-0336-585X

Reviewer #1 (Public review): https://doi.org/10.7554/eLife.99003.3.sa1
Reviewer #2 (Public review): https://doi.org/10.7554/eLife.99003.3.sa2
Author response https://doi.org/10.7554/eLife.99003.3.sa3

## Additional files

### Supplementary files

MDAR checklist

### Data availability

The data that support the findings of this article are openly available: https://doi.org/10.5281/zenodo.17449659.

The following dataset was generated:

| Author(s) | Year | Dataset title | Dataset URL | Database and Identifier |
|---|---|---|---|---|
| Costello J, Kanso E | 2025 | Dataset - Feeding rates in sessile versus motile ciliates are hydrodynamically equivalent | https://doi.org/10.5281/zenodo.17449659 | Zenodo, 10.5281/zenodo.17449659 |

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

## Appendix 1

### 1 Collection of published biological data

#### 1.1 Kinematic measurements

We collected from published literature measurement data of microorganisms, including size, swimming speed of motile organisms, and surrounding flow speed of sessile organisms. For modeling and further analyzing, we simplified the shape of those organisms as a sphere with equivalent sphere diameters. Based on each organism's coarse morphology (*Appendix 1—figure 1A*), we first approximated these morphologies into regular shapes (*Appendix 1—figure 1B*), then, by calculating the volume of those shapes, we solved for the radius $a$ of a model sphere that has the same volume. The data is included in the attached Excel sheet.

Using the raw data, we plot $U/a$ in *Appendix 1—figure 2A* in regular scale and in *Appendix 1—figure 2B* in logarithmic scale using the equivalent spherical radius $a$ as the length scale, the swimming speed as the velocity scale $U$ for the motile organism, and the maximum observed flow speed as the velocity scale $U$ for sessile organisms. In addition, we plot $Ua$ in *Appendix 1—figure 2C*, representing advection strength.

#### 1.2 Phylogenetic information

For each species for which we collected morphometric and flow kinematic data, we traced it back along the evolutionary tree. The diatoms, swimming, and sessile ciliates were traced back to their common eukaryotic ancestor. From that common ancestor, various clades evolved (*Grattepanche et al., 2018*; *Hinchliff et al., 2015*; *Keeling et al., 2005*; *Parfrey et al., 2008*), including the supergroup SAR from which ciliates descended, and the Viridiplantae group to which green algae like the volvox belong. We combined the taxonomy information of the species collected above to obtain the tree of life diagram in the main text.

### 2 Mathematical modeling

#### 2.1 Stokes equations

At the microscopic scale, the Reynolds number Re, which quantifies the ratio of inertial to viscous forces is nearly zero. The viscous forces are dominant, and the fluid velocity $\mathbf{u}$ is governed by the incompressible Stokes equation,

$$-\nabla p + \eta \nabla^2 \mathbf{u} = 0, \qquad \nabla \cdot \mathbf{u} = 0, \tag{1}$$

where $p$ is the pressure field and $\eta$ is the dynamic viscosity.

#### 2.2 Stokeslet model

Following (*Andersen and Kiørboe, 2020*; *Dölger et al., 2017*), we represented the ciliary activity as a point force $\mathbf{F} = -F_{\text{cilia}}\mathbf{e}_z$ located at $L\mathbf{e}_z$, at a distance $L$ from the center of the sphere (*Appendix 1—figure 3*). For the motile sphere, the flow field is obtained by superposition of two solutions of the Stokes equation, to obtain a solution with no slip boundary condition at the sphere's surface: $\mathbf{u} = \mathbf{u}_1 + \mathbf{u}_2$, where $\mathbf{u}_1 = \mathcal{G}_o \cdot \mathbf{F}$ and $\mathcal{G}_o$ is Green's tensor due to a point force near a rigid sphere (*Blake, 1971a*; *Kim and Karrila, 1991*) and $\mathbf{u}_2$ is the solution due to uniform flow $-U\mathbf{e}_z$ pass a rigid sphere, expressed in the sphere's frame of reference (*Andersen and Kiørboe, 2020*; *Kim and Karrila, 1991*).

In the Stokes regime, force balance on the swimming sphere is given by

$$\mathbf{0} = \mathbf{T} + \mathbf{K} + \mathbf{D}, \tag{2}$$

where $\mathbf{T} = -\mathbf{F}$ is the thrust force generated by the flagella (Stokeslet) in the swimming direction $\mathbf{e}_z$, $\mathbf{D}$ is the drag force due to a moving sphere in fluid with speed $U$, and $\mathbf{K}$ is the hydrodynamic force acting on the sphere by the flow generated by the point force $\mathbf{F}$. Evaluating these terms leads to a scalar equation in the $\mathbf{e}_z$-direction that can be solved to obtain the swimming speed

$$U = \frac{F_{\text{cilia}}}{6\pi\mu a}\left(1 - \frac{3a}{2L} + \frac{a^3}{2L^3}\right). \tag{3}$$

For a sessile sphere, the force balance on the sphere is given by:

$$\mathbf{0} = \mathbf{F}_{\text{tether}} + \mathbf{T} + \mathbf{K}, \tag{4}$$

where $\mathbf{T} = -\mathbf{F}$ and $\mathbf{K}$ are defined as above. This equation can be solved to obtain the force $\mathbf{F}_{\text{tether}}$ provided by a tether to prevent the sphere from moving in the $\mathbf{e}_z$ direction. Namely,

$$\mathbf{F}_{\text{tether}} = \left( 1 - \frac{3a}{2L} + \frac{a^3}{2L^3} \right) F_{\text{cilia}} \mathbf{e}_z, \tag{5}$$

To non-dimensionalize these solutions in *Equation 3; Equation 5*, we chose $a$ as the characteristic length scale and $\mathcal{U} = F_{\text{cilia}}/(8\pi\mu a)$ as the characteristic velocity scale. Results based on our implementation are shown in *Appendix 1—figure 3* and are consistent with the results of *Andersen and Kiørboe, 2020*.

## 2.3 Envelope model

We reproduced the general solution of *Equation 1* subject to arbitrary slip velocity at the surface of the sphere and proper decay at infinity. The solution is given in terms of the radial distance $r$ from the center of the sphere and an angular variable $\mu = \cos\theta$, in the form of an expansion in Legendre polynomials $P_n(\mu)$(*Blake, 1971b*; *Liu et al., 2025*; *Michelin and Lauga, 2010*).

We considered the 'treadmill' slip velocity at the surface of the sphere

$$u_\theta\big|_{r=a} = B\sqrt{1-\mu^2}, \qquad u_r\big|_{r=a} = 0, \tag{6}$$

where $B$ is a constant parameter that defines a velocity scale $\mathcal{U}$. In this case, the solution simplifies significantly. In *Appendix 1—table 1*, we list mathematical expressions for the boundary conditions, fluid velocity field, pressure field, forces acting on the sphere, hydrodynamic power, and swimming speed for freely moving ciliated sphere. Results based on our analysis are consistent with the results of *Michelin and Lauga, 2010* for a swimming sphere.

## 2.4 Clearance rate

The clearance rate is calculated by integrating the $z$-component of the flow velocity $u_z$ past an annular disk around the equator of the sphere (*Appendix 1—figure 3*). Normalizing the result by a flux $\pi R^2 U$ of a uniform flow $U$ through a disk of area $\pi R^2$, we get the clearance rate (with minus sign to indicate positive clearance value)

$$Q = -\frac{2}{R^2 U} \int_a^R u_z\big|_{z=0}\, r\, dr. \tag{7}$$

When applied to the Stokeslet model, this clearance rate is consistent with that in *Andersen and Kiørboe, 2020*; *Figure 3*. By applying it to the fluid solution in the envelope model (*Appendix 1—table 1*), we can compare feeding rates across the two models ciliary models, which reflect different distributions of the ciliary force – concentrated in the Stokeslet model versus fully distributed in the envelope model.

### Advection-diffusion equation

To determine the effect of the advective currents generated by the attached and freely-swimming ciliated sphere on the concentration distribution around the sphere, we considered the advection-diffusion equation for the steady-state concentration $C$ of nutrients around the spherical surface

$$\mathbf{u} \cdot \nabla C = D\Delta C. \tag{8}$$

Here, $\mathbf{u} \cdot \nabla C$ and $D\Delta C$ are, respectively, the advective and diffusive rates of change of the nutrient concentration field $C$. Considering the absorption of nutrients is limited by the number of surface receptors, we assume nutrient concentration on sphere surface is constant (*Berg and Purcell, 1977*; *Bialek, 2012*). Thus, by removing the constant from the background concentration, the concentration at the spherical surface can be considered zero (*Magar et al., 2003*; *Michelin*

*and Lauga, 2011*). The boundary conditions, for both the attached and free-swimming spheres, are given by

$$C(\mu)\big|_{r=a=1} = 0, \quad C(\mu)\big|_{r\to\infty} = C_\infty. \tag{9}$$

For non-dimensionalization, we scale the concentration using $c = (C_\infty - C)/C_\infty$ assuming zero concentration at an infinite distance and uniform concentration on the sphere's surface. By choosing characteristic velocity scale $\mathcal{U}$ and length scale $a$, we obtain the dimensionless governing equation and boundary conditions.

$$\mathrm{Pe}\,\mathbf{u}\cdot\nabla c = \nabla^2, \quad c(r=1) = 1, \quad c(r\to\infty) = 0, \tag{10}$$

where $\mathrm{Pe} = a\mathcal{U}/D$.

## Sherwood number

By Fick's law, a gradient in concentration yields a flux. The nutrient uptake rate is the area integral of the flux over the spherical surface $I = -\oint \hat{\mathbf{n}}\cdot(-D\nabla C)\mathrm{d}S$, where $\mathrm{d}S = 2\pi R^2 \sin\theta\,\mathrm{d}\theta$ is the element of surface area of the sphere. This sign convention is such that the concentration flux is positive if the sphere takes up nutrients. In the absence of microcurrents, the concentration is governed by diffusion only. The steady-state concentration obtained by solving the diffusion equation $\nabla^2 c = 0$ internally bounded by a sphere is given by $c(r) = 1/r$ in non-dimensional form; for which, $C(r) = C_\infty(1 - a/r)$. The steady-state inward current due to molecular diffusion is given by $I_{\mathrm{diffusion}} = 4\pi a D C_\infty$.

For diffusion coupled with advective microcurrents, the nutrient uptake is quantified by the Sherwood number Sh, which is equivalent to a dimensionless nutrient uptake, where $I$ is scaled by $I_{\mathrm{diffusion}}$. The dimensionless form of the Sherwood number is

$$\mathrm{Sh} = -\frac{1}{2}\int_{-1}^{1} \nabla c \cdot \mathbf{e}_r\big|_{r=1}\,d\mu. \tag{11}$$

*Equation 11* provides an alternative metric for evaluating nutrient uptake to the one in *Equation 7*. By applying *Equations 7; Equation 11* to the Stokeslet and envelope models, we are able to compare nutrient uptake across two ciliary models that reflect different distributions of ciliary forces – concentrated versus fully distributed force – using two different metrics. Having two metrics – flow rate *Equation 7* and Sherwood number *Equation 11* – allows us to test the robustness of the results to the choice of metric. For example, when applied to the Stokeslet model, *Equation 11* allows us to test the robustness of the results in *Andersen and Kiørboe, 2020* to the choice of metric, concluding that their results are not universal.

## Numerical methods

Given the symmetry of all fluid fields under consideration, we solve for the corresponding concentration fields in spherical coordinates $(r, \theta, \phi)$ with $\phi$ axis-symmetry. To solve for the Stokeslet model, we use the finite difference method on a two-dimensional mesh $(r, \theta)$ that is stretched to ensure a denser grid near the sphere and a sparser grid further away (*Fletcher, 2012*). And for solving the envelope model, we use the Legendre spectral method (*Michelin and Lauga, 2011*), which uses Legendre polynomials as the spectrum basis in $\theta$ dimension and finite difference in $r$ dimension with a stretched mesh.

## 2.6 Asymptotic analysis

For analyzing the advection-diffusion equation under extreme Péclet number $\mathrm{Pe} \ll 1$ and $\mathrm{Pe} \gg 1$, we use the approach employed in *Acrivos and Goddard, 1965*; *Acrivos and Taylor, 1962* for a rigid sphere and in *Magar et al., 2003* for the spherical envelope model.

## Small Pe

At small Péclet number, we expand the concentration field as

$$c = \mathrm{Pe}^0 c_0 + \mathrm{Pe}^1 c_1 + \mathrm{Pe}^2 c_2 + \ldots \tag{12}$$

We substitute the expanded concentration into the dimensionless advection-diffusion *Equation 10*, we arrive at a system of equations associated with each order in Pe, $\mathrm{Pe}^0$, $\mathrm{Pe}^1$, $\mathrm{Pe}^2$. At the leading order $\mathrm{Pe}^0$, the solution is simply $c_0 = 1/r$. To find the solution at higher orders, we substitute the velocity field corresponding to each model (see *Appendix 1—table 1*) and solve for the higher order equations. We solve for the equations associated with the first three orders of Pe, the asymptotic solution to $\mathrm{Pe} \ll 1$ for all considered models are in *Appendix 1—table 3*.

### Large Pe

For $\mathrm{Pe} \gg 1$, we take the Taylor series expansion of flow field at $r = 1$ and keep only the leading terms

$$u_r(r,\mu) = u_r\big|_{r=1} + \frac{\partial u_r}{\partial r}\Big|_{r=1}(r-1) + \dots, \qquad u_\theta(r,\mu) = u_\theta\big|_{r=1} + \frac{\partial u_\theta}{\partial r}\Big|_{r=1}(r-1) + \dots,$$

$$= -2(r-1)\mu + \dots, \qquad\qquad = \sqrt{1-\mu^2} + \dots. \tag{13}$$

We define the temporary variable $y = r - 1$ (not to be confused with the $y$-coordinated in the inertial $(x, y, z)$ space). The region $y \ll 1$ represents a thin boundary layer around the spherical surface. Since the concentration boundary layer is expected to be thinner as Pe increases, we rescale $r - 1 = y = \mathrm{Pe}^{-m}Y$, where $Y$ is a new variable. By substituting the new variable and the linearized flow field *Equation 13* into the advection-diffusion equation *Equation 8*, and matching order of Pe on both sides of the advection-diffusion equation, we obtain $m = 1/2$ for envelope model and $m = 1/3$ for rigid sphere model. By substituting $m$ with keeping only the leading order, we can transfer the partial differential equation into an ordinary equation by inducing a new similarity variable (*Magar et al., 2003*). By solving the ODEs, we obtain the asymptotic solutions to $\mathrm{Pe} \gg 1$ for all models listed in *Appendix 1—table 3*.

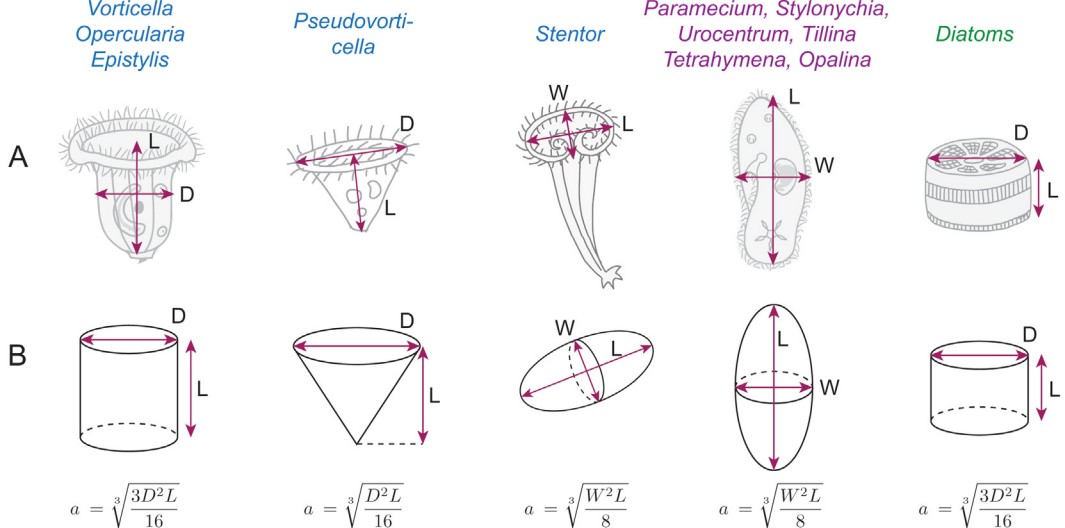

**Appendix 1—figure 1.** Geometry of ciliates and diatoms. (**A**) Representative morphologies of surveyed species of sessile ciliates (blue), swimming ciliates (purple), and sinking diatoms (green). (**B**) Simplified shapes of above organisms. The volumes of those shapes can be obtained as $V_{\mathrm{cylinder}} = \pi D^2 L/4, V_{\mathrm{cone}} = \pi D^2 L/12$, and $V_{\mathrm{spheroid}} = \pi W^2 L/6$. Later on, we further simplify those shapes to an equal volume-based sphere with radius $a$, calculated as the formula shown in B for each geometry.

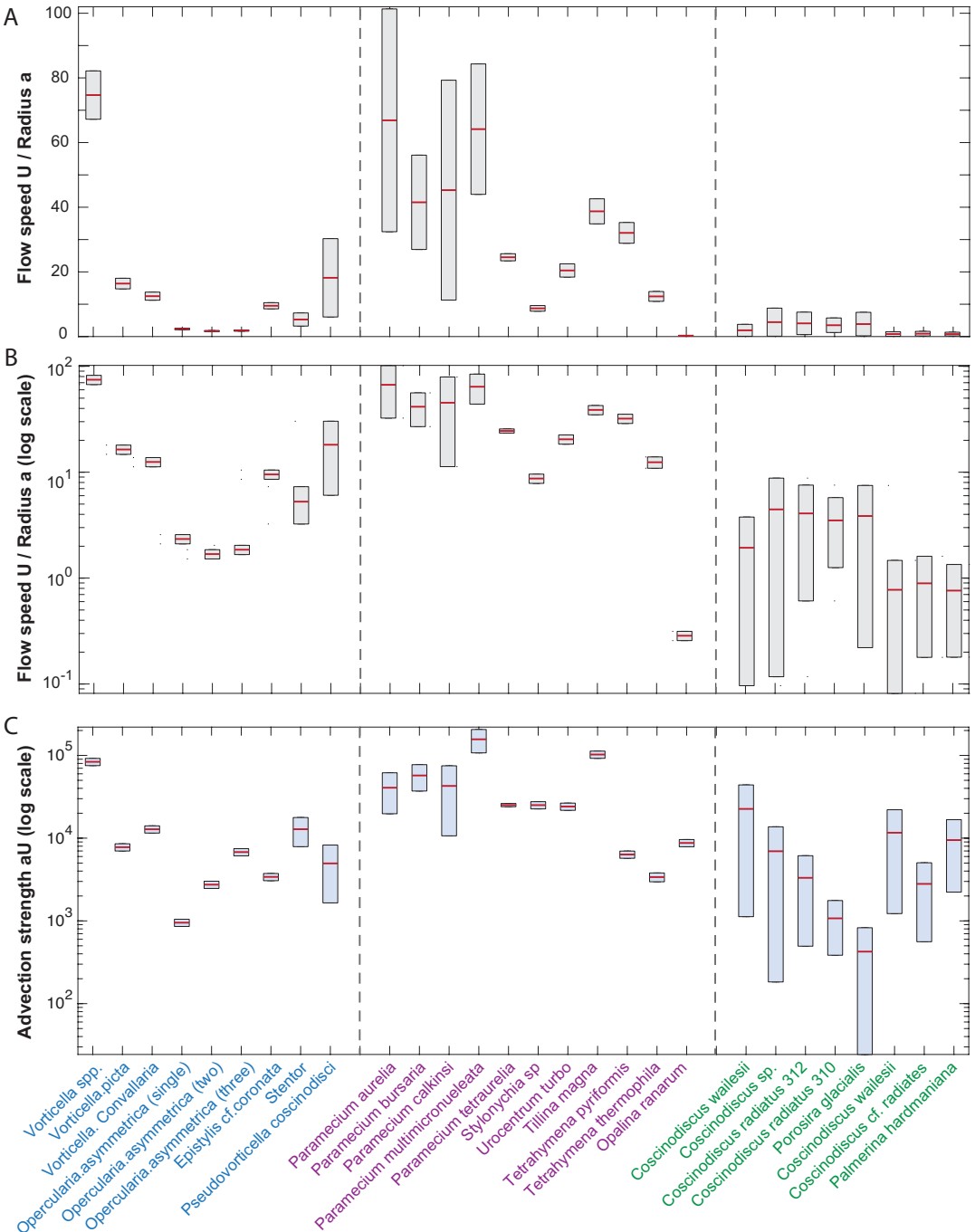

**Appendix 1—figure 2.** Raw data of character flow speed and size for sessile (blue) and swimming (purple) ciliates, and sinking diatoms (green). For motile organisms, the characteristic flow speed $U$ is the swimming speed, while for sessile organisms, the characteristic speed $U$ is the maximum flow speed reported near the organism. The characteristic length $a$ is based on the volume-equivalent spherical radius. (**A**) Ratio of flow speed to length scale $U/a$ in regular scale. (**B**) Ratio of flow speed to length scale $U/a$ in logarithmic scale. (**C**) Advection strength $aU$ in logarithmic scale.

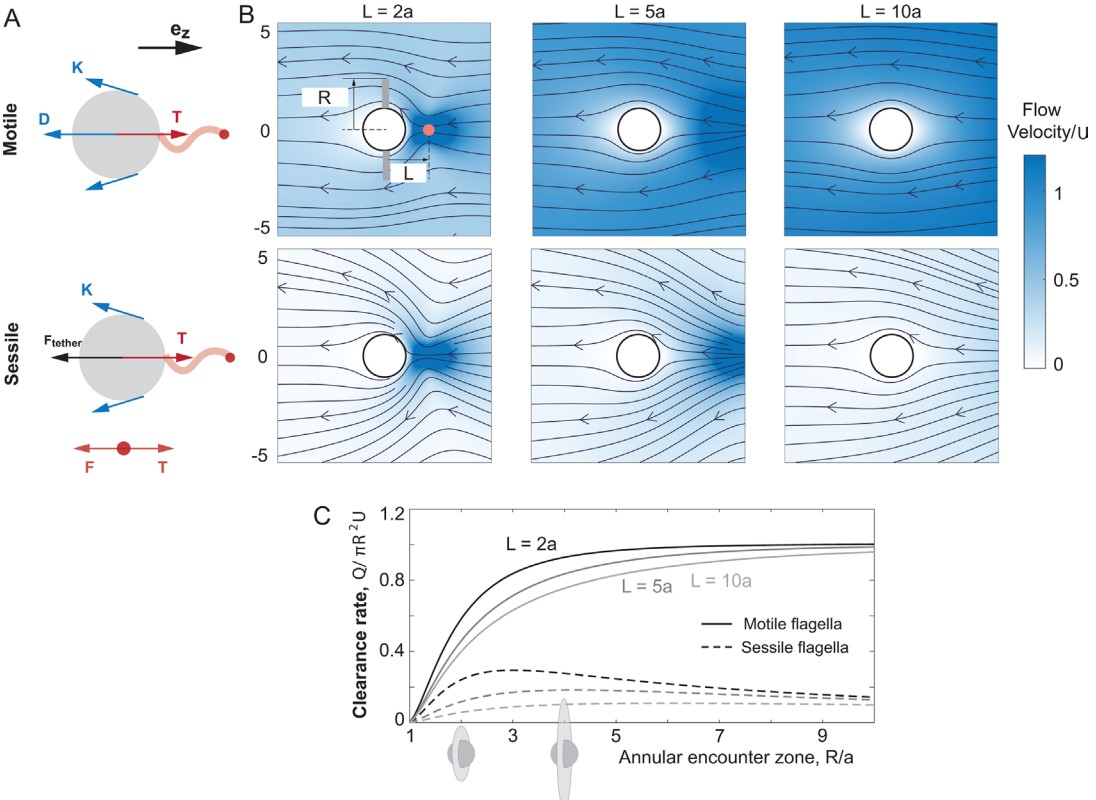

**Appendix 1—figure 3.** Stokeslet model. (**A**) Flow field around sessile and swimming spheres. Solution in non-dimensional form for $a = 1$ and $B = F_{\text{cilia}}/(8\pi\mu a) = 1$. (**B**) Fluid around a motile and sessile sphere with same point force strength at distance $L = [2a, 5a, 10a]$, respectively. (**C**) Normalized clearance varies as annular encounter disk $R$ for above same three point force distances. $L = [2a, 5a, 10a]$

**Appendix 1—table 1.** Envelope model subject to treadmill slip velocity.

Mathematical expressions of boundary conditions, fluid velocity field, pressure field, forces acting on the sphere, hydrodynamic power, and speed for each representative case: sessile ciliated sphere, freely swimming ciliated sphere, and sinking (non-ciliated) sphere. All quantities are given in dimensional form in terms of the radial distance $r$ and angular variable $\mu = \cos\theta$.

| | B.C. at the surface of the sphere | B.C. at infinity |
|---|---|---|
| Sessile ciliated sphere | $u_\theta\big|_{r=a} = B\sqrt{1-\mu^2},\ u_r\big|_{r=a} = 0$ | $\mathbf{u}\big|_{r\to\infty} = \mathbf{0}$ |
| Swimming ciliated sphere | $u_\theta\big|_{r=a} = B\sqrt{1-\mu^2},\ u_r\big|_{r=a} = 0$ | $\mathbf{u}\big|_{r\to\infty} = -U\mathbf{e}_z$ |
| Sinking (non-ciliated) sphere | $\mathbf{u}\big|_{r=a} = 0$ | $\mathbf{u}\big|_{r\to\infty} = -U\mathbf{e}_y$ |

| | Fluid velocity field |
|---|---|
| Sessile | $u_r(r,\mu) = \left(\dfrac{a^3}{r^3} - \dfrac{a}{r}\right)B\mu,\qquad u_\theta(r,\mu) = \dfrac{1}{2}\left(\dfrac{a^3}{r^3} + \dfrac{a}{r}\right)B\sqrt{1-\mu^2}$ |
| Swimming | $u_r(r,\mu) = \left(-\dfrac{2}{3} + \dfrac{2a^3}{3r^3}\right)B\mu,\quad u_\theta(r,\mu) = \left(\dfrac{2}{3} + \dfrac{a^3}{3r^3}\right)B\sqrt{1-\mu^2},$ |
| Sinking | $u_r(r,\mu) = \left(-1 + \dfrac{3a}{2r} - \dfrac{a^3}{2r^3}\right)U\mu,\quad u_\theta(r,\mu) = \left(1 - \dfrac{3a}{4r} - \dfrac{a^3}{4r^3}\right)U\sqrt{1-\mu^2}$ |

| | Fluid velocity field in lab frame | Far-field signature |
|---|---|---|
| Sessile | same as above | Force monopole ($u \sim 1/r$) (Stokeslet) |
| Swimming | $u_r(r,\mu) = \dfrac{2a^3}{3r^3}B\mu,\ u_\theta(r,\mu) = \dfrac{a^3}{3r^3}B\sqrt{1-\mu^2}$ | Potential dipole ($u \sim 1/r^3$) |
| Sinking | $u_r(r,\mu) = \left(\dfrac{3a}{2r} - \dfrac{a^3}{2r^3}\right)U\mu,$ $u_\theta(r,\mu) = \left(-\dfrac{3a}{4r} - \dfrac{a^3}{4r^3}\right)U\sqrt{1-\mu^2}$ | Force monopole ($u \sim 1/r$) (Stokeslet) |

**Appendix 1—table 2.** Appendix 1—table 1 extension.

| | Fluid pressure field | Forces on sphere |
|---|---|---|
| Sessile | $p(r,\mu) = p_\infty - \eta\dfrac{a}{r^2}B\mu$ | $\mathbf{F} = 4\pi\eta aB\mathbf{e}_z$ |
| Swimming | $p(r,\mu) = p_\infty - \eta\dfrac{a}{r^2}B\mu$ | $\mathbf{F} = 4\pi\eta aB\mathbf{e}_z,\ \mathbf{D} = -6\pi\eta aU\mathbf{e}_z$ |
| Sinking | $p(r,\mu) = p_\infty + \eta\dfrac{3a}{2r^2}U\mu$ | $\mathbf{W} = -\dfrac{4}{3}\pi a^3(\delta\rho)g\mathbf{e}_y,\ \mathbf{D} = 6\pi\eta aU\mathbf{e}_y$ |

| | Hydrodynamic power | Swimming speed |
|---|---|---|
| Sessile | $\mathcal{P} = 8\pi a\eta B^2$ | $U = 0$ |
| Swimming | $\mathcal{P} = 8\pi a\eta\dfrac{2}{3}B^2$ | $U = \dfrac{2}{3}B$ |
| Sinking | $\mathcal{P} = 6\pi a\eta U^2$ | $U = \dfrac{2ga^2(\delta\rho)}{9\eta}$ |

**Appendix 1—table 3.** Expressions for Sh as a function of Pe for sessile and swimming ciliated sphere model, compared to a sinking sphere.

| | Large Pe limit | | Small Pe limit | |
|---|---|---|---|---|
| | Sherwood number | Reference | Sherwood number | Reference |
| Sessile | $\mathrm{Sh} = \dfrac{2}{\sqrt{3\pi}}\mathrm{Pe}^{\frac{1}{2}}$ | Present study | $\mathrm{Sh} = 1 + \dfrac{43}{720}\mathrm{Pe}^2$ | Present study |
| Swimming | $\mathrm{Sh} = \dfrac{2}{\sqrt{3\pi}}\mathrm{Pe}^{\frac{1}{2}}$ | *Magar et al., 2003*; *Michelin and Lauga, 2011* | $\mathrm{Sh} = 1 + \dfrac{1}{3}\mathrm{Pe}$ | *Magar et al., 2003*; *Michelin and Lauga, 2011* |
| Sinking | $\mathrm{Sh} = 0.55\mathrm{Pe}^{\frac{1}{3}}$ | *Acrivos and Goddard, 1965*; *Acrivos and Taylor, 1962*; *Guasto et al., 2012* | $\mathrm{Sh} = 1 + \dfrac{1}{3}\mathrm{Pe}$ | *Acrivos and Taylor, 1962*; *Guasto et al., 2012* |

