## [Editor Report · eLife Assessment]

This **important** paper addresses the role of fluid flows in nutrient uptake by microorganisms propelled by the action of cilia or flagella. Using a range of mathematical models for the flows created by such appendages, the authors provide **convincing** evidence that the two strategies of swimming and sessile motion can be competitive. These results will have significant implications for our understanding of the evolution of multicellularity in its various forms.

---

## [Referee Report · Reviewer #1 (Public review)]

Summary:

The manuscript studies nutrient intake rates for stationary and motile microorganisms to assess the effectiveness of swim vs. stay strategies. This work provides valuable insights on how the different strategies perform in the context of a simplified mathematical model that couples hydrodynamics to nutrient advection and diffusion. The swim and stay strategies are shown to yield similar nutrient flux under a range of conditions.

Strengths:

Strengths of the work include (i) the model prediction in Fig. 3 of nutrient flux applied to a range of microorganisms including an entire clade that are known to use different feeding strategies and (ii) a study of the interaction between cilia and absorption coverage showing the robustness of their predictions provided these regions have sufficient overlap.

Weaknesses:

In the revision, the authors have adequately addressed the weaknesses I raised in the first round of review.

---

## [Referee Report · Reviewer #2 (Public review)]

Summary:

The authors have collected a significant amount of data from the literature on the flow regimes associated with microorganisms whose propulsion is achieved through the action of cilia or flagella, with particular interest in the competition between sessile and motile lifestyles. They then use several distinct hydrodynamic models for the cilia-driven flows to quantify the nutrient uptake and clearance rate, reported as a function of the Peclet number. Among the interesting conclusions the authors draw concerns the question of whether, for certain ciliates, there is a clear difference in nutrient uptake rates in the sessile versus motile forms. The authors show that this is not the case, thereby suggesting that the evolutionary pressure associated with such a difference is not present. The analysis also includes numerical calculations of the uptake rate for spherical swimmers in the regime of large Peclet numbers, where the authors note an enhancement due to advection-generated thinning of the solutal boundary layer around the organism.

Strengths:

In addressing the whole range of organism sizes and Peclet numbers the authors have achieved an important broad perspective on the problem of nutrient uptake of ciliates, with implications for understanding evolutionary driving forces toward particular lifestyles (e.g. sessile versus motile).

---

## [Author Response]

The following is the authors’ response to the original reviews.

**Reviewer #1 (Public Review):**
Summary:The manuscript studies nutrient intake rates for stationary and motile microorganisms to assess the effectiveness of swim vs. stay strategies. This work provides valuable insights on how the different strategies perform in the context of a simplified mathematical model that couples hydrodynamics to nutrient advection and diffusion. The swim and stay strategies are shown to yield similar nutrient flux under a range of conditions.Strengths:Strengths of the work include (i) the model prediction in Fig. 3 of nutrient flux applied to a range of microorganisms including an entire clade that are known to use different feeding strategies and (ii) a study of the interaction between cilia and absorption coverage showing the robustness of their predictions provided these regions have sufficient overlap.

We thank the referee for their thorough review of our manuscript and for their constructive feedback.

Weaknesses: To improve the work, the authors should further expand their discussion of the following points:(1) The authors comment that a number of species alternate between sessile and motile behavior. It would be helpful to discuss what is known about what causes switching between these modes and whether this provides insights regarding the advantages of the different behaviors.

The transition between sessile and motile states is often influenced by external environmental conditions, such as prey availability and predator presence, which determine the most advantageous state at any given time. For instance, members of the genus *Stentor* are known to detach from their colonies and exhibit solitary swimming behavior in response to low prey abundance (Tartar, 2013) or when avoiding predators (Dexter et al. 2019). Similarly, the transition in *Vorticella* is influenced by chemical cues, such as pH (Baufer et al., 1999) or algae concentration (Langlois, 1975).

References:

Dexter, J. P., Prabakaran, S., & Gunawardena, J. (2019). A complex hierarchy of avoidance behaviors in a single-cell eukaryote. Current biology, 29(24), 4323-4329.

Tartar, V. (2013). The biology of stentor: International series of monographs on pure and applied biology: Zoology. Elsevier.

BAUFER, P. J. D., Amin, A. A., Pak, S. C., & BUHSE JR, H. E. (1999). A method for the synchronous induction of large numbers of telotrochs in Vorticella convallaria by monocalcium phosphate at low pH. Journal of Eukaryotic Microbiology, 46(1), 12-16.

LANGLOIS, G. A. (1975). Effect of algal exudates on substratum selection by motile telotrochs of the marine peritrich ciliate Vorticella marina. The Journal of Protozoology, 22(1), 115-123.

(2) An encounter zone of R=1.1a appears be used throughout the manuscript, but I could not find a biological justification for this particular value. This results appear to be quite sensitive to this choice, as shown in Supplement Fig. 3(B). In the Discussion, it is mentioned that using a much larger exclusion zone leads to significantly different nutrient flux, and it is implied that such a large exclusion zone is not biologically plausible, but this was not explained sufficiently.

Thank you for pointing this out. We chose the value of the encounter zone based on a rough calculation of cilia length relative to body length. Cilia are typically of the order of 10 microns in length, and the cell body of a ciliate is typically of the order of 100-1000 microns.

For example, in the work of Jiang, H., & Buskey, E. J., 2024, I&II, the nutrient encounter is reported at the leading edge of the ciliary band in *Strombidium* and *Amphorides*. Here, cilia appear to be about 20% of the body length and the particles are absorbed quite close to the cell surface. A similar encounter near the cell surface is reported in Gilmour, 1978 and Thomazo et al., 2020.

In the theoretical model of Andersen and Kiørboe (2020), a much larger encounter zone, extending 10 times the body length (that is, an encounter zone that is 1000% larger than the body length). This is obviously not biologically justifiable.

We edited the manuscript to better justify our choices and provide supporting references.

References:

Andersen, A., & Kiørboe, T. (2020). The effect of tethering on the clearance rate of suspension-feeding plankton. Proceedings of the National Academy of Sciences, 117(48), 30101-30103.

Jiang, H., & Buskey, E. J. (2024). Relating ciliary propulsion morphology and flow to particle acquisition in marine planktonic ciliates II: the oligotrich ciliate Strombidium capitatum. Journal of Plankton Research, fbae011.

Jiang, H., & Buskey, E. J. (2024). Relating ciliary propulsion morphology and flow to particle acquisition in marine planktonic ciliates I: the tintinnid ciliate Amphorides quadrilineata. Journal of Plankton Research, fbae012.

Gilmour, T. H. J. (1978). Ciliation and function of the food-collecting and waste-rejecting organs of lophophorates. Canadian Journal of Zoology, 56(10), 2142-2155.

Thomazo, J. B., Le Révérend, B., Pontani, L. L., Prevost, A. M., & Wandersman, E. (2021). A bending fluctuation-based mechanism for particle detection by ciliated structures. Proceedings of the National Academy of Sciences, 118(31), e2020402118.

(3) In schematic of the in Fig. 2(B) it was unclear if the encounter zone in the envelope model is defined analogously to the Stokeslet model or if a different formulation is used.

Yes, we defined the encounter zone the same in both models. In fact, we used two metrics for evaluating nutrient uptake: one considers only the fluid flow rate through an encounter zone, another considers the mass transport within the fluid and absorption at the entire ciliary surface. For the first metric, the clearance rate Q, evaluated by calculating the flow rate past an annular disk, it is consistent applied to all models, depicted in Figure 2(B). The second metric, the nutrient uptake rate, which we define as the dimensionless integration of mass flux over the entire spherical surface, is also consistently applied to all models to evaluate Sh number. Both metrics are evaluated on the Stokeslet and envelope models.

We edited the main text to further clarify these two metrics in the revision.

(4) The force balance argument should be clarified. Equation (3) of the supplement gives the force-velocity relation in the motile case. Since equation (4), which the authors state is the net force in the sessile case, seems to involve the same expression, would it not follow from U=0 in the sessile case that one would simply obtain quiescent flow with Fcilia = 0?

The force balance equations for the model organism differ between the motile and sessile modes. In the submitted version, SI Eq.(3) and SI Eq.(4) are derived from different force balance equations, where the velocity U does not appear in the sessile Stokeslet model.

For the Stokeslet model, the force generated by the flagella acting on the fluid is modeled as a point force \begin{document}$\mathbf{F}=-\mathrm{F}_{\mathrm{cilia}} \boldsymbol{e}_{z}$\end{document}

Motile Stokeslet model:

The force balance on the sphere is given by: \begin{document}$\mathbf{0}=\mathbf{T}+\mathbf{K}+\mathbf{D}$\end{document}

Where \begin{document}$\mathbf{T}=\mathrm{F}_{\text {cilia }} \boldsymbol{e}_{z}$\end{document} is the thrust force generated by the flagella in the direction of swimming, \begin{document}$\mathbf{D}=-6 \eta \pi a U \boldsymbol{e}_{2}$\end{document} is the drag force due to a moving sphere in fluid with speed U, and K is the hydrodynamic force acting on the sphere by the flow generated by the point force F. For a given strength of the Stokeslet, \begin{document}$\mathbf{F}=-\mathrm{F}_{\text {cilia }} \boldsymbol{e}_{z}$\end{document}, the swimming speed U can be calculated by the force balance.

Sessile Stokeslet model:

The force balance on the sphere is given by: \begin{document}$\mathbf{0}=\mathbf{F}_{\text {tether }}+\mathbf{T}+\mathbf{K}$\end{document}

Where , T = -F, and K are defined as above. Similarly, for a given point force F, the required force provided by a stalk to fix the sphere can be calculated by the force balance.

Therefore, SI Eq.(3) and (4), are not directly applicable across both the Stokeslet and envelope models. While the expressions appear similar due to the presence of the forces F and K, separate calculations are needed depending on the force model.

We edited the SI document and SI Figure 3 to clarify this.

Reference:

Andersen, A., & Kiørboe, T. (2020). The effect of tethering on the clearance rate of suspension-feeding plankton. Proceedings of the National Academy of Sciences, 117(48), 30101-30103.

**Reviewer #2 (Public Review):**
Summary:The authors have collected a significant amount of data from the literature on the flow regimes associated with microorganisms whose propulsion is achieved through the action of cilia or flagella, with particular interest in the competition between sessile and motile lifestyles. They then use several distinct hydrodynamic models for the cilia-driven flows to quantify the nutrient uptake and clearance rate, reported as a function of the Peclet number. Among the interesting conclusions the authors draw concerns the question of whether, for certain ciliates, there is a clear difference in nutrient uptake rates in the sessile versus motile forms. The authors show that this is not the case, thereby suggesting that the evolutionary pressure associated with such a difference is not present. The analysis also includes numerical calculations of the uptake rate for spherical swimmers in the regime of large Peclet numbers, where the authors note an enhancement due to advection-generated thinning of the solutal boundary layer around the organism.Strengths:In addressing the whole range of organism sizes and Peclet numbers the authors have achieved an important broad perspective on the problem of nutrient uptake of ciliates, with implications for understanding evolutionary driving forces toward particular lifestyles (e.g. sessile versus motile).

We thank the referee for their thorough review of our manuscript and for their feedback regarding the inclusion of more relevant references.

Weaknesses:The authors appear to be unaware of rather similar calculations that were done some years ago in the context of Volvox, in which the issue of the boundary layer size and nutrient uptake enhancement were clearly recognized [M.B. Short, et al., Flows Driven by Flagella of Multicellular Organisms Enhance Long-Range Molecular Transport, PNAS 103, 8315-8319 (2006)]. This reference also introduced the model of a fixed shear stress at the surface of the sphere as a representation of the action of the cilia, which may be more realistic than the squirmer-type boundary condition, although the two lead to similar large-Pe scalings.

We apologize for having missed to include this reference in the submitted version of the manuscript. We read this work thoroughly, it is indeed highly relevant to the present study.

The findings reported in Figure 4, that the uptake rate is robust to variations in cilia coverage and absorption fraction, are similar in spirit to an observation made recently in the context of the somatic cell neighbourhood areas in Vovox [Day, et al., eLife 11, e72707 (2022)]. There, it was found that while there is a broad distribution of those areas, and hence of the coarse-grained tangential flagellar force acting on the fluid, the propulsion speed is rather insensitive to those variations.

Thank you for pointing us to the work of Day, et al., eLife 11, e72707 (2022). We did not know about this study and have not read it before. The work is broadly relevant to our study, and we added a reference to this work in the discussion.